# Safe and effective thrombolysis in free flap salvage: Intra-arterial urokinase infusion

**Jun Hyeok Kim**[1], **Sungyeon Yoon**[1], **Heeyeon Kwon**[2], **Deuk Young Oh**[1], **Young-Joon Jun**[1], **Suk-Ho Moon**[1]*

**1** Department of Plastic & Reconstructive Surgery, College of Medicine, The Catholic University of Korea, Seoul, Republic of Korea, **2** Banobagi Plastic Surgery Clinic, Seoul, Republic of Korea

* nasuko@catholic.ac.kr

**Data Availability Statement:** All relevant data are within the paper and its Supporting information files.

## Abstract

Despite the high success rate in reconstruction using free tissue transfer, flap failure is often caused by microvascular thrombosis. In a small percentage of cases with complete flap loss, a salvage procedure is performed. In the present study, the effectiveness of intra-arterial urokinase infusion through the free flap tissue was investigated to develop a protocol to prevent thrombotic failure. The retrospective study evaluated the medical records of patients who underwent salvage procedure with intra-arterial urokinase infusion after reconstruction with free flap transfer between January 2013 and July 2019. Thrombolysis with urokinase infusion was administered as salvage treatment for patients who experienced flap compromise more than 24 hours after free flap surgery. Because of an external venous drainage through the resected vein, 100,000 IU of urokinase was infused into the arterial pedicle only into the flap circulation. A total of 16 patients was included in the present study. The mean time to re-exploration was 45.4 hours (range: 24–88 hours), and the mean quantity of infused urokinase was 69,688 IU (range: 30,000–100,000 IU). 5 cases presented with both arterial and venous thrombosis, while 10 cases had only venous thrombosis and 1 case had only arterial thrombosis; in a study of 16 patients undergoing flap surgery, 11 flaps were found to have survived completely, while 2 flaps experienced transient partial necrosis and 3 were lost despite salvage efforts. In other word, 81.3% (13 of 16) of flaps survived. Systemic complications, including gastrointestinal bleeding, hematemesis, and hemorrhagic stroke, were not observed. The free flap can be effectively and safely salvaged without systemic hemorrhagic complications using high-dose intra-arterial urokinase infusion within a short period of time without systemic circulation, even in delayed salvage cases. Urokinase infusion results in successful salvage and low rate of fat necrosis.

## Introduction

Reconstruction using microvascular free tissue transfer is reliable and versatile, with > 95% success rate due to advances in microsurgical technique and postoperative management [1–7]. However, despite a high success rate, free flap failure is often caused by microvascular and peri-anastomotic thrombosis [8–10], and thrombotic failure can occur even when the

**Funding:** This work was supported by the Korea Medical Device Development Fund grant funded by the Korea, government (Ministry of Science and ICT; Ministry of Trade, Industry, and Energy; Ministry of Health & Welfare; Ministry of Food and Drug Safety; Project Number: 202012E02). Recieved by SH Moon. The funders had no role in study design, data collection and analysis, decision to publish, or preparation of the manuscript.

**Competing interests:** The authors have declared that no competing interests exist.

procedure is performed by experienced microvascular surgeons [4, 7]. The incidence rate of thrombosis in free flap is 10–15%, and approximately 75% are salvaged through re-exploration [5, 6].

Re-exploration is performed in 6–14% of free flap transfers and is associated with a small percentage of complete flap loss [11, 12]. If vascular thrombosis is confirmed, systemic heparin is administered, and thrombectomy or thrombolysis is performed [13]. The three thrombolytic agents commonly used in microvascular surgery are streptokinase, urokinase, and tissue plasminogen activator (TPA) [3, 10]. The thrombolytic agents convert plasminogen to plasmin, which then dissolves intraluminal thrombi [14]. Among the agents, urokinase has a shorter half-life and greater efficacy than streptokinase [3, 10].

The present study investigates the effectiveness of high-dose intra-arterial urokinase infusion within a short period of time without systemic circulation through free flap tissue to develop a protocol for re-exploration to prevent thrombotic failure.

## Methods

This single-center retrospective study evaluated the medical records of patients who underwent salvage procedures with intra-arterial urokinase infusion after reconstruction with free flap transfer between January 2013 and July 2019. 581 cases of free flap transfer were performed during the study period.

In cases where flap compromise occurred more than 24 hours after free flap surgery, thrombolysis with urokinase infusion was administered for salvage treatment regardless of the presence or absence of arterial thrombus, in the presence of venous thrombus or venous insufficiency. In that case, both arterial reliance and the congestive flap was confirmed in re-exploration more than 24 hours postoperatively, and intra-flap thrombosis was suspected due to decreased venous drainage, so it was necessary to actively perform thrombolysis in a way that has less systemic side effects through external venous drainage while using thrombolytic agents at high concentrations.

The time to re-exploration, type of anesthesia, site of thrombus, quantity of infused urokinase, venous drainage after salvage, and flap survival were evaluated. Furthermore, systemic complications due to urokinase, including hematoma formation, gastrointestinal bleeding, hematemesis, and hemorrhagic stroke, were observed during postoperative follow-up.

Our study was approved by the Institutional Review Board (Catholic Medical Center Office of Human Research Protection Program): KC22RISI0303. And the requirement for informed consent was waived due to the retrospective nature of the study.

### Thrombolysis protocol

First, the thrombi filled along the vein were removed by venotomy or extracted by heparin irrigation through the resected end of the vein. Next, in a patient with thrombus in the artery, the arterial thrombus was removed by direct thrombectomy, and the surgeon confirmed that arterial pulsation and blood flow were restored after the arterial anastomosis. 100,000 IU urokinase mixed with 100 mL normal saline was continuously infused into the arterial pedicle with a butterfly needle regardless of the front and the rear of anastomosis site (Fig 1). To prevent mixed urokinase from entering systemic circulation, injecting site was after the branch of the main artery, and the external veinous drainage was maintained. The urokinase bag mixed with saline was compressed by a pressure infusion bag up to 250 mmHg to overcome arterial pressure. Simultaneously, the venous flow was discharged outside the blood vessel through the resected end to prevent urokinase infusion into systemic circulation. When urokinase reached the inside of the flap and a thrombolysis occurred, bleeding or fluid mixed with urokinase was

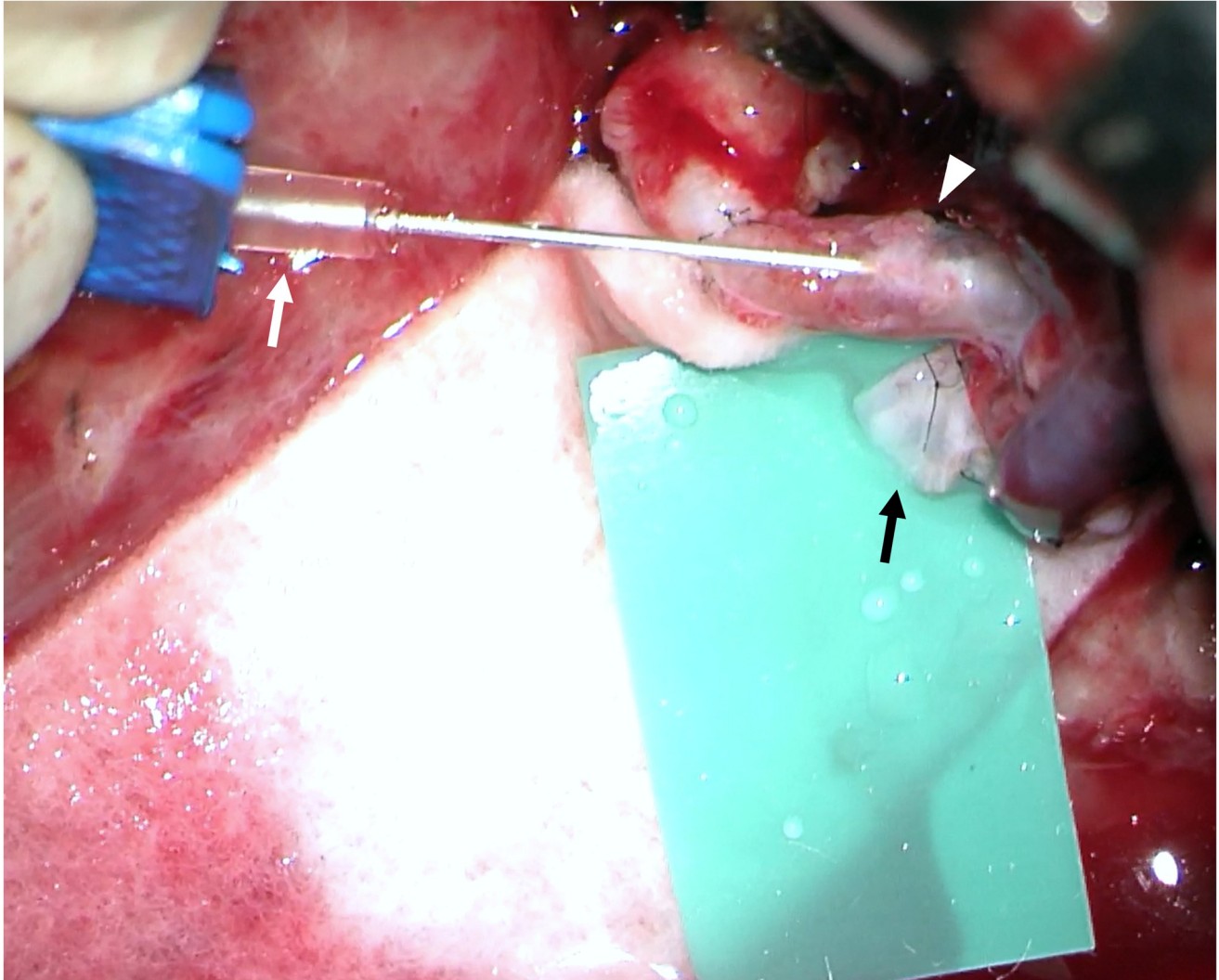

**Fig 1. Urokinase infused into an arterial pedicle of flap.**

observed at the margin of the flap. Eventually, a transparent fluid draining through the venous pedicle was observed. When venous drainage was observed at a sufficient rate, urokinase infusion was stopped, and venous drainage was continued for 15 minutes at the venotomy or venous pedicle based on the half-life of urokinase. (S1 Video) If the venous drain was insufficient, the above-mentioned process was repeated by reinfusing urokinase. Then, venotomy repair or venous anastomosis was performed, and the flap and extension incision site were closed without compression of the pedicle. For postoperative management, heparin was administered to prevent recurrent thrombus formation for 7 days and adjusted while monitoring activated partial thromboplastin time (aPTT) range of 30 to 50 seconds 3 times a day.

The flap was meticulously monitored for any clinical signs of vascular compromise, including color, temperature, capillary refill rate, skin tension, turgor, and Doppler sound, every hour for the first 24 hours, every 2 hours for up to 48 hours, and every 4 hours for up to 120 hours postoperatively. In addition, portable ultrasonography was used to evaluate the blood flow of pedicles, which provided an objective assessment of the pedicle's perfusion and enabled

early detection of potential thrombotic changes that may not be evident on clinical examination or Doppler ultrasound alone.

White arrow indicates a butterfly needle, white arrowhead indicates the arterial pedicle of flap, black arrow indicates venous pedicle; semi-transparent fluid leaked from the venotomy site.

## Statistical analyses

For continuous variables, the mean and standard deviation were used for description and the difference between groups was compared using Mann-Whitney test. For nominal variables, fractions in percentages were calculated and chi-square test was used for comparison. A p-value < 0.05 indicated a statistically significant difference.

## Results

A total of 16 patients (10 males and 6 females) was included in the present study, and the mean age was 59.7 years (range: 15–76 years). The demographics, cause of defect, location of defect, and type of flap are shown in Table 1. (S1 Table) The mean time from initial operation to re-exploration was 45.4 hours (range: 24–88 hours), and the mean quantity of infused urokinase was 69,688 IU (range: 30,000–100,000 IU). The type of anesthesia in re-exploration was general anesthesia in 9 patients and local anesthesia in 7 patients. 6 cases had arterial thrombosis and 15 cases had venous thrombosis. To clarify, 5 cases presented with both arterial and venous thrombosis, while 10 cases had only venous thrombosis and 1 case had only arterial thrombosis. Venous drainage after salvage procedure was observed in 14 cases; in a study of 16 patients undergoing flap surgery, 11 flaps were found to have survived completely, while 2

**Table 1. Patient demographics.**

| Variables | Values |
|---|---|
| **Sex** | |
| Male | 10 (62.5%) |
| Female | 6 (37.5%) |
| **Age (yrs)** | 59.7 ± 14.8 (15–76) |
| **Defect cause** | |
| Cancer | 6 (37.5%) |
| Trauma | 4 (25.0%) |
| DM foot | 3 (18.8%) |
| Chronic ulcer | 2 (12.5%) |
| Radiation ulcer | 1 (6.3%) |
| **Defect region** | |
| Low extremity | 10 (62.5%) |
| Head and neck | 5 (31.3%) |
| Breast | 1 (6.3%) |
| **Flap type** | |
| ALT | 12 (75.0%) |
| TDAP | 2 (12.5%) |
| DIEP | 1 (6.3%) |
| Fibular OC | 1 (6.3%) |

Abbreviations: DM, diabetes mellitus; ALT, anterolateral thigh; TDAP, thoracodorsal artery perforator; DIEP, deep inferior epigastric perforator; OC, osteocutaneous

**Table 2. Surgical details.**

| Variables | Values |
|---|---|
| **Time to re-exploration (hrs)** | 45.4 ± 18.1 (24–88) |
| **Type of Anesthesia** | |
| General | 9 (56.3%) |
| Local | 7 (43.8%) |
| **Thrombose in artery** | |
| (+) | 6 (37.5%) |
| (-) | 10 (62.5%) |
| **Thrombose in vein** | |
| (+) | 15 (93.8%) |
| (-) | 1 (6.3%) |
| **Urokinase infusion (IU)** | 69,688 ± 25,395 (30,000–100,000) |
| **Venous drainage after salvage** | |
| (+) | 14 (87.5%) |
| (-) | 2 (12.5%) |
| **Flap Survival** | |
| (+) | 13 (81.3%) |
| (-) | 3 (18.8%) |

flaps experienced transient partial necrosis and 3 were lost despite salvage efforts. In other word, 81.3% (13 of 16) of flaps survived. (Table 2). In addition, difference between groups with successful results or failure was not statistically significant in sites of thrombotic formation, time to re-exploration, and quantity of urokinase infusion (Table 3). Systemic complications, including gastrointestinal bleeding, hematemesis, or hemorrhagic stroke, were not observed in any patient; however, 2 cases showed hematoma at the donor site.

## Case 1

A 60-year-old male patient underwent wide resection of a malignant melanoma lesion in the left heel area and reconstruction with a free anterolateral thigh flap (Fig 2A). Immediately after surgery, the flap appeared fine; however, flap congestion progressed with time, and flap salvage procedure was performed 65 hours after the operation (Fig 2B). Both pedicle artery and vein were thrombosed. After infusion of 100,000 IU urokinase, vigorous venous drainage was observed, and the flap was less congested (Fig 2C). At postoperative 2 months, the flap survived well without further secondary procedures (Fig 2D).

**Table 3. Difference between flap survival and total loss groups.**

| Variables | Flap survival | | p-value |
|---|---|---|---|
| | (+) | (-) | |
| **Site of thrombus** | | | 0.0889 |
| Veins | 9 (56.3%) | 1 (6.3%) | |
| Artery | 0 (0.0%) | 1 (6.3%) | |
| Both artery and vein | 4 (25.0%) | 1 (6.3%) | |
| **Time to re-exploration (hrs)** | 45.4 ± 19.5 (24–88) | 45.7 ± 13.7 (31–58) | 0.6875 |
| **Urokinase infusion (IU)** | 66,538 ± 26,723 (30,000–100,000) | 83,333 ± 14,434 (75,000–100,000) | 0.4161 |

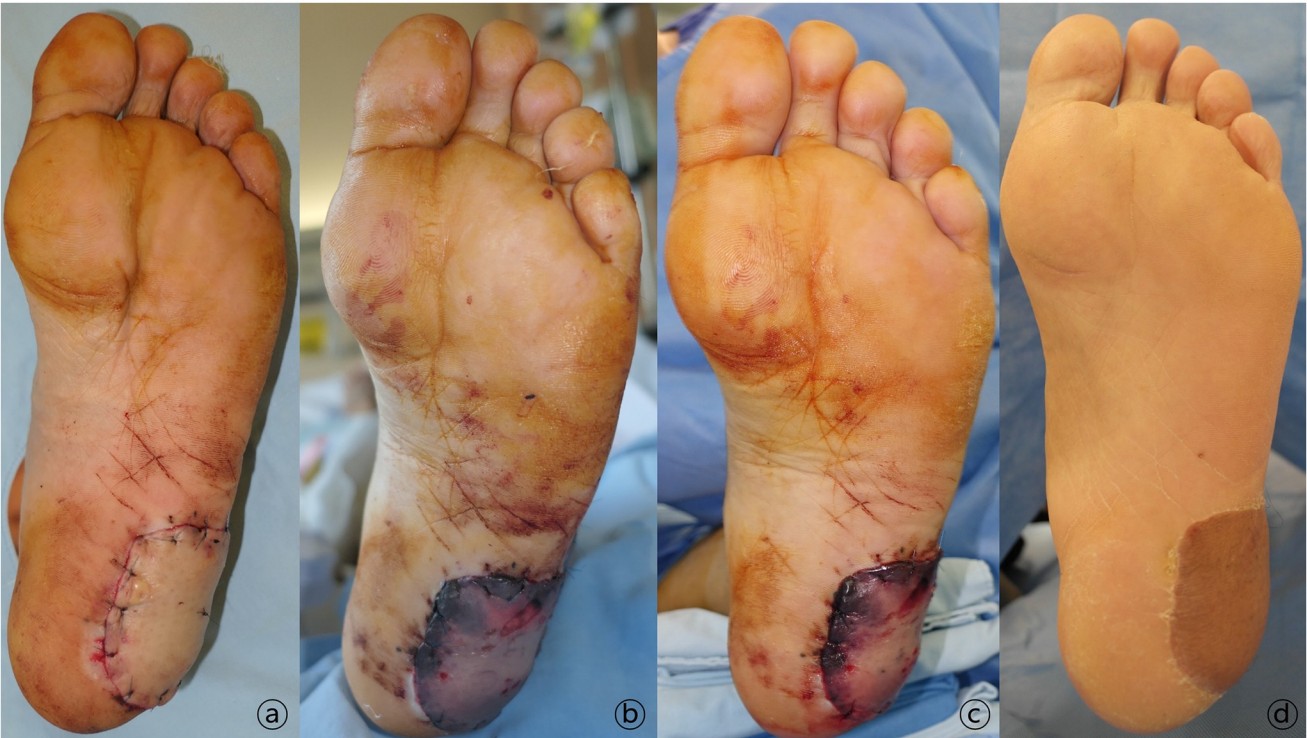

**Fig 2. A 60-year-old male patient who underwent wide resection of a malignant melanoma in the left heel area and reconstruction with a free anterolateral thigh flap.** A. Clinical photograph immediately after surgery showing a good flap that progressed to congestion over time. B. Both pedicle artery and vein were thrombosed 65 hours after the surgery. After 100,000 IU of urokinase were infused, vigorous venous drainage was observed. C. Photograph immediately after flap salvage procedure. D. Clinical photograph 3 months postoperatively without further secondary procedures.

### Case 2

A 57-year-old female patient underwent debridement of a radiation ulcer in her left foot insole area and reconstruction with a free anterolateral thigh flap (Fig 3A). Immediately after surgery, the flap appeared fine; however, flap congestion progressed (Fig 3B). Flap salvage procedure was performed 48 hours after the surgery (Fig 3C). The vein was clotted with thrombi; however, the arterial pedicle appeared normal. After infusion of 75,000 IU urokinase, vigorous venous drainage was observed. At postoperative 18 months, the surviving flap was well-maintained (Fig 3D).

## Discussion

Intra-arterial urokinase infusion is an effective micro-thrombolytic strategy in delayed salvage cases, resulting in 81.3% (13 of 16) successful salvage and total flap survival with partial necrosis and secondary healing in two cases. Furthermore, systemic hemorrhagic complications do not occur when using a thrombolytic agent because the external venous drainage from the flap pedicle prevents urokinase inflow into systemic circulation.

In this study, pharmacological thrombolysis with urokinase was considered in the case of venous thrombus or venous insufficiency even if it was not thrombosis, regardless of the presence or absence of arterial thrombus in the revision performed 24 hours after flap surgery when venous congestion signs had occurred such as purplish flap, shortened blood refill time less than three seconds, venous bleeding at pin prick or the edge of flap, or edematous flap. At that time, both arterial reliance and the congestive flap was confirmed in re-exploration more

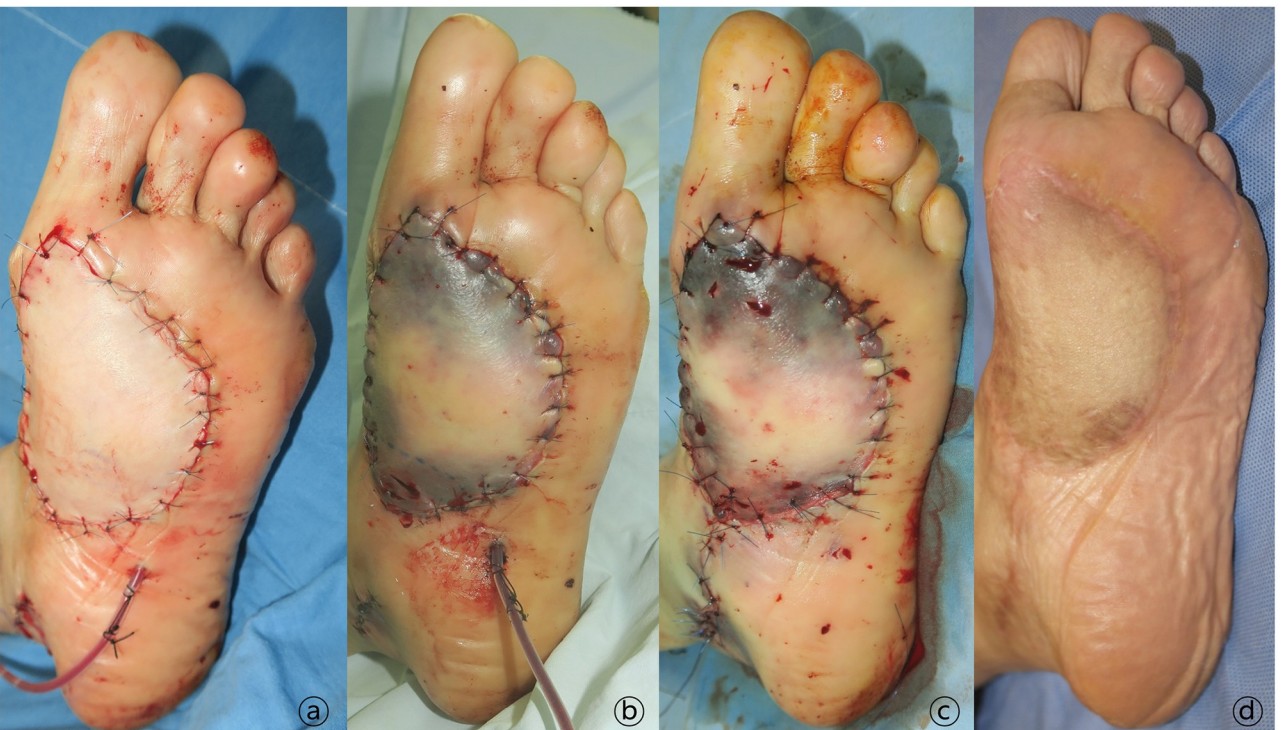

**Fig 3. A 57-year-old female patient who underwent debridement of a radiation ulcer in her left foot insole area and reconstruction with a free anterolateral thigh flap.** A. Clinical photograph immediately after surgery showing a good flap that progressed to congestion over time. B. The vein was clotted with thrombi 48 hours after the surgery. After 75,000 IU of urokinase were infused, vigorous venous drainage was observed. C. Photograph immediately after flap salvage procedure. D. The surviving flap was well-maintained 18 months postoperatively.

than 24 hours postoperatively. Because the decreased venous drainage and congestive clinical signs may suggest intra-flap thrombosis, it was required to aggressively initiate thrombolysis with less systemic complications through external drainage while using thrombolytic agents at high dose.

Intra-arterial administration of urokinase with external drainage has been demonstrated to be an effective method for delivering excessive doses of the thrombolytic agent within a short time frame. This approach offers the advantage of bypassing systemic circulation, reducing the risk of systemic hemorrhagic complications. In free flap surgery, where extensive bleeding can occur at both the donor and recipient sites, the use of thrombolytic agents is crucial to minimize the risk of bleeding-related complications and improve surgical outcomes. In this study, the infusion rate of urokinase has been reached up to 600,000 IU per hour, with a typical infusion time of 15 minutes. While high doses of urokinase can be effective in treating thromboembolic disorders, it is important to note that doses exceeding 500,000 IU per day may increase the risk of systemic adverse effects, including intracranial hemorrhage, hematochezia, and gross hematuria. Ref.14.

Despite the reconstruction by microvascular free tissue transfer has become a safe and reliable with greatly improved success rates [1–7], flap failure occurs infrequently by microvascular and peri-anastomotic thrombosis [8–10] even performed by an experienced micro-vascular surgeon [4, 7]. Thrombosis occurs in 10–15% of free flap transfers, of which 75% is salvaged through re-exploration [5, 6]. Usually, arterial thrombosis is caused due to technical errors during anastomosis and can be resolved by repeat arterial anastomosis. Whereas the cause of

venous thrombosis is unclear [15]. but it occurs three times more than arterial thrombosis [16].

Flap salvage methods include anticoagulation, thrombolysis, and thrombectomy. Anticoagulants such as heparin, enoxaparin, and aspirin, have been used by 96% of reconstructive surgeons in free flap surgery [17, 18]. Heparin inactivates the coagulation cascade [19] and decreases the formation of fibrin [20, 21]. The antithrombotic effect of heparin is measured by monitoring activated partial thromboplastin time. However, heparin is associated with a risk of hemorrhage-forming hematoma and heparin-induced thrombocytopenia [22, 23]. Aspirin inhibits platelet aggregation and impairs thrombin generation [24]. The platelet dysfunction caused by aspirin is also associated with risk of hemorrhage, which requires additional transfusion and re-operation [25].

Three thrombolytic agents commonly used in microvascular surgery are streptokinase, urokinase, and TPA [3, 10]. The thrombolytic agents convert plasminogen to plasmin, which then dissolve intraluminal thrombi [14]. Among the agents, urokinase has s shorter half-life (15 minutes) and greater efficacy than streptokinase [3, 10]. Urokinase shows superior improvement in patency than the combination of anticoagulants including heparin and dextran [26]; however, the thrombolytic agent also increases the risk of hemorrhage [7].

Comparing the three agents, it has been reported that there is no difference in effect among the three thrombolytic agents [3, 27, 28]. However, urokinase may have several advantages over streptokinase, because urokinase has less antigenicity, can be infused in high concentrations by directly activating plasminogen, and has less systemic side effect due to its shorter half-life [3]. TPA can resist microcirculation thrombosis by interfering with leukocyte-endothelial interactions [28] and theoretically has less systemic hemorrhage rate, although clinical evidence is insufficient [3].

Thrombectomy includes direct thrombectomy and conventional thrombectomy. In direct thrombectomy, a side branch is clipped near the pedicle thrombus or an additional transverse incision is made to approach the full extent of thrombus [29]. In conventional thrombectomy, a Fogarty catheter (#2 or #3 French) is passed through the anastomosis opening and retrieved after ballooning until the prominent thrombus is removed [30]. However, because thrombectomy cannot extract microthrombi within the flap, medical agents are required for distant microthrombi.

In comparison of urokinase thrombolysis with mechanical thrombectomy, this study showed 81.3% survival rate (13 of 16 flaps). On the other hand, three studies that performed mechanical thrombectomy as a salvage procedure result flap survival rate of 20.0% (1 of 4 flaps) [31], 55.6% (5 of 9 flaps) [32], and 60% (18 of 30 flaps) [29], respectively. From the pooled analysis of three studies, 24 out of 43 flaps underwent mechanical thrombectomy were salvaged with survival rate of 55.8%. It is lower than the survival rate from the present study, then thrombolysis with urokinase demonstrates the potentially beneficial effect as flap salvage procedures compared to mechanical thrombectomy.

The pathogenesis of venous thrombosis is different from arterial thrombosis. Venous thrombosis consists of fibrin clotting and arterial thrombosis of platelet aggregation [33]. Similar to the results of the present study, fibrinolysis was more important in microvascular occlusion than anticoagulation of platelet aggregation because venous thrombosis is more likely to occur than arterial thrombosis in the thrombotic failure of free tissue transfer [5]. Therefore, the thrombolytic agent urokinase is more efficient for delayed flap salvage with either venous thrombosis or venous obstruction along the flap congestion than is the combination of anticoagulants.

Gentle manipulation of vessels during micro-anastomosis is crucial to prevent damage to the endothelium and minimize the risk of thrombosis. In our institute, we use a meticulous

technique that involves the use of stay sutures placed at 180 degrees along the vessel circumference, with anterior suturing performed first. To assist with optimal exposure, a cottonoid or similar gauze material with adequate frictional force is placed under the blood vessel and used to carefully rotate the vessel by 180 degrees. Hydrostatic dilation is achieved by squirting heparinized saline solution into the vessel lumen under pressure, which provides adequate exposure of the posterior wall while continuously monitoring the correct positioning of the stitches. This method avoids the need for an Ikuta approximator, which further minimizes the risk of injury to the delicate vascular tissue.

Hemodynamics play a crucial role in vascular compromise during free flap surgery, and while hemodilution has been shown to improve tissue oxygenation due to accelerated erythrocyte velocity with reduced blood viscosity [34] in ischemic flaps with better microcirculation [35] and high flap survival rates [36] in animal model. In addition, normovolemic hemodilution has been found to be a considerably effective method for reducing the incidence of thrombosis and microvascular free flap failure [37].

The present study had several limitations including the retrospective design and a relatively small number of patients without a control group. And, a selection bias may have been present in our study, as the ALT flap was the most commonly utilized flap in our institution and thus represented the majority of flaps included in the analysis.

Unlike in previous studies in which thrombolytic agents were infused into systemic circulation, in the present study, urokinase was injected only into the flap circulation, minimizing the amount absorbed into the body to reduce systemic complications caused by the thrombolytic agent. Furthermore, urokinase injection is an effective method with a relatively high success rate of revision more than 24 hours after free flap surgery.

## Conclusion

The free flap can be effectively and safely salvaged, even in delayed salvage cases more than 24 hours after primary surgery, using high-dose intra-arterial urokinase infusion within a short period of time without systemic circulation resulting in the avoidance of systemic hemorrhagic complications and results in successful salvage and total flap survival.

## Supporting information

**S1 Video. The process of thrombolysis by infusing urokinase into the arterial pedicle of the flap.**
(MP4)

**S1 Table. Raw data of this study.**
(XLSX)

## Author Contributions

**Conceptualization:** Jun Hyeok Kim, Suk-Ho Moon.

**Data curation:** Jun Hyeok Kim.

**Formal analysis:** Sungyeon Yoon.

**Funding acquisition:** Suk-Ho Moon.

**Investigation:** Sungyeon Yoon, Heeyeon Kwon.

**Methodology:** Sungyeon Yoon, Heeyeon Kwon.

**Project administration:** Heeyeon Kwon, Suk-Ho Moon.

**Resources:** Heeyeon Kwon.

**Software:** Jun Hyeok Kim, Heeyeon Kwon.

**Supervision:** Deuk Young Oh, Young-Joon Jun.

**Validation:** Deuk Young Oh, Young-Joon Jun.

**Visualization:** Jun Hyeok Kim.

**Writing – original draft:** Jun Hyeok Kim.

**Writing – review & editing:** Suk-Ho Moon.

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
