## [Decision Letter · Decision Letter 0]

21 Nov 2022

PONE-D-22-28593Safe and Effective Thrombolysis in Free Flap Salvage: Intra-arterial Urokinase InfusionPLOS ONE

Dear Dr. Moon,

Thank you for submitting your manuscript to PLOS ONE. After careful consideration, we feel that it has merit but does not fully meet PLOS ONE’s publication criteria as it currently stands. Therefore, we invite you to submit a revised version of the manuscript that addresses the points raised during the review process.

We look forward to receiving your revised manuscript.

Kind regards,

Fabio Santanelli, di Pompeo d'Illasi, MD, PhD

Academic Editor

PLOS ONE

Journal Requirements:

"No"

5. Please include a copy of Tables 1 to 4 which you refer to in your text.

Reviewers' comments:

Reviewer's Responses to Questions

**Comments to the Author**

1. Is the manuscript technically sound, and do the data support the conclusions?

Reviewer #1: Yes

Reviewer #2: Partly

2. Has the statistical analysis been performed appropriately and rigorously? 

Reviewer #1: Yes

Reviewer #2: N/A

3. Have the authors made all data underlying the findings in their manuscript fully available?

Reviewer #1: Yes

Reviewer #2: No

4. Is the manuscript presented in an intelligible fashion and written in standard English?

Reviewer #1: Yes

Reviewer #2: Yes

5. Review Comments to the Author

Reviewer #1: The study is complete and presented in an intelligible fashion. Data are consistent and results are correct. Statistical analysis has been provided in a correct way and also the surgical technique was accurate. We are perfectly agreeing to publication.

Reviewer #2: While your manuscript is interesting and presents potential, it presents some serious concerns which must be accounted for and addressed accordingly.

• None of the tables from 1 through 4 appear to be present in the current version of your submission. This is a big oversight: the data is crucial and needs to be assessed.

• In Serletti’s study from the 1990s (ref. 14), they salvaged flaps with venous compromise by administering 250.000 IU of urokinase into the arterial recipient vessel proximally to the arterial anastomosis through the free flap, prior to 5000 IU of heparin administered intravenously, with no heparin post-operatively. They reported 8 free flaps with vascular compromise: 3 had evidence of arterial thrombosis, while 5 had venous compromise. They tested their protocol on the latter 5 cases. The aim of your study is to investigate “the effectiveness of intra-arterial urokinase infusion through free flap tissue to develop a protocol for re-exploration to prevent thrombotic failure.” However, the rationale for your study is not very clear: your study performed over two decades later presents a slightly higher patient population of free flap salvage cases (16 vs 5). Nevertheless, the differences which made your study design unique are not addressed sufficiently. Namely, the external venous drainage through the resected vein, the pressure infusion pump, the lower urokinase dosage. Was your aim the validation of a thrombolytic protocol for free flap salvage with strategies to reduce the risk of hemorrhagic events? Please clarify this in your abstract and at the end of your introduction.

• Your protocol was used just for the cases with venous compromise, which was 15 cases, regardless of whether arterial compromise was also present concomitantly or not (6 had arterial thrombosis). You state that 13 of 16 (81.3%) flaps survived completely, and just developed 2 partial necrosis? That’s 15. What happened to the 16th flap? The way you present your figures is very confusing. Just state how many out of the starting 581 flaps developed vascular compromise, how many of those were of venous origin and required salvage using your protocol.

• In your “Thrombolysis protocol” you mention that urokinase infusion was stopped “after the vein was sufficiently drained”. Could you define that with more precision? Was it enough to observe transparent fluid draining through the venous pedicle?

• While Serletti did not administer any heparin post-operatively, you did so to prevent recurrent thrombus formation. Was the dosage adjusted according to patient weight? What was the ideal aPTT range you were seeking? How long is post-operative anticoagulation after free flap salvage maintained and is it correlated with what you routinely perform in your facility for systemic thromboembolic prophylaxis?

• You stated that your mean “time to re-exploration was 45.4 hours (range: 24–88 hours)”. As you know, Serletti et al. (ref. 14) conducted a study similar to yours in 1998. He used no flap monitoring strategy other than an external Doppler probe and clinical assessment. They reported a mean re-exploration time of 3.6 days. How would you explain the significant difference? Did you use any flap-monitoring technique?

• Evidence has shown that the rate of flap salvage significantly decreased after 48 h. (PMID: 28648581) Were there any cases of late venous thrombosis (defined as beyond 72 hours)? There should be since the higher range of your mean time to re-exploration was 88 hours, but this could not be assessed due to the missing tables. If so, were they managed differently? Were the complications you reported correlated with time to re-exploration?

• In Serletti’s study, out of the 5 salvaged flaps, 2 developed marginal loss of tissue while one had some fat necrosis. That’s a 60% flap-related complication rate due to the prolonged ischemia. Why would you say that you only had 2 cases of partial flap necrosis while 13 flaps survived completely (i.e. 18.4% flap-related complications). How do you explain this difference?

• In case of potential endothelial or mechanical damage from vessel handling, could you clarify which micro-anastomosis technique was used? Was is the posterior wall-first approach? Was heparinized solution used during the procedure? (PMID: 24706545)

• The flaps used in reconstructive surgery are prone to ischemia and hypoxia, which imply a considerable risk of wound-healing complications. Hemodilution has been shown to improve tissue oxygenation in ischemic flaps. (PMID: 12170063) Was hemodilution taken into consideration when planning the flap? (PMID: 26313824; PMID: 26818324)

• Regarding your discussion, consider addressing more substantially the many differences with other studies such as Serletti et a. (ref. 14) and Nelson et al (ref. 16) compared to yours. Additionally, it should be stated how selective administration of thrombolytic agents to the free flap’s arterial circulation at lower dosages (100.000 IU) has already been shown in anecdotal evidence to prevent systemic complications, despite not having an external venous drainage. (PMID: 2810204) Additionally, your limitations section should address how future research should strive to obtain larger cohorts in the setting of a clinical trial to make this evidence more robust.

• The conclusions are not very satisfying: your protocol aims at making free flap salvage safer by reducing the overall urokinase dosage and by avoiding systemic circulation unlike what has been proposed in previous protocols. This should be stressed appropriately.

• Just as a minor observation: while the contents are very understandable, there are a few English language mistakes spread in the manuscript (i.e. “Intra-arterial urokinase infusion is an effective micro-thrombolysis in delayed salvage cases” in the first sentence of your Discussion: do you mean “an effective thrombolytic agent” or “thrombolytic strategy”?). I would recommend that the manuscript is spell-checked by a native speaker.

6. PLOS authors have the option to publish the peer review history of their article (what does this mean?). If published, this will include your full peer review and any attached files.

Reviewer #1: No

Reviewer #2: **Yes: **Guido Firmani

---

## [Author Response · Author response to Decision Letter 0]

15 Dec 2022

We appreciate your comments and request to correct the manuscript. We believe that the valuable comments from the reviewers helped to improve our manuscript. We hope that the paper is now suitable for publication in PLOS one.

Reviewer #1: The study is complete and presented in an intelligible fashion. Data are consistent and results are correct. Statistical analysis has been provided in a correct way and also the surgical technique was accurate. We are perfectly agreeing to publication.

ANSWER: We appreciate your kind comments.

Reviewer #2: While your manuscript is interesting and presents potential, it presents some serious concerns which must be accounted for and addressed accordingly.

• None of the tables from 1 through 4 appear to be present in the current version of your submission. This is a big oversight: the data is crucial and needs to be assessed.

ANSWER: We agree with you. Unfortunately, four tables were omitted while editing the manuscript to meet the format of PLOS one. We added them appropriately to the revised manuscript. 

• In Serletti’s study from the 1990s (ref. 14), they salvaged flaps with venous compromise by administering 250.000 IU of urokinase into the arterial recipient vessel proximally to the arterial anastomosis through the free flap, prior to 5000 IU of heparin administered intravenously, with no heparin post-operatively. They reported 8 free flaps with vascular compromise: 3 had evidence of arterial thrombosis, while 5 had venous compromise. They tested their protocol on the latter 5 cases. The aim of your study is to investigate “the effectiveness of intra-arterial urokinase infusion through free flap tissue to develop a protocol for re-exploration to prevent thrombotic failure.” However, the rationale for your study is not very clear: your study performed over two decades later presents a slightly higher patient population of free flap salvage cases (16 vs 5). Nevertheless, the differences which made your study design unique are not addressed sufficiently. Namely, the external venous drainage through the resected vein, the pressure infusion pump, the lower urokinase dosage. Was your aim the validation of a thrombolytic protocol for free flap salvage with strategies to reduce the risk of hemorrhagic events? Please clarify this in your abstract and at the end of your introduction.

ANSWER: We appreciate your comment. As you know, if venous thrombosis occurs in the free flap, even if revision surgery is performed, the loss of flap, including partial necrosis, would be inevitable. Microsurgeons are all burdened with this catastrophic event, and the suggestion of a safe thrombolysis method would be welcome for all. It is helpful to use the urokinase like this study, but the expression of the systemic side effect using urokinase has been an obstacle to its use. This is because free flap surgery has many bleeding sites on the donor and recipient sites, so the use of verified heparin has always been a subject to pay attention. Therefore, as in this study, it is a great advantage that high-dose urokinase can be used in a short period of time without systemic circulation. You said that our urokinase is less used, but if we calculate our dose per hour, it is upto 600,000 IU/hr because infusion time is about 10 minutes. The use of more than 500,000 IU of urokinase per day may have a systemic side effect such as intracranial hemorrhage, hematochezia, and gross hematuria. Although Serletti did not mention side effects, the lack of a protocol for the urokinase since 1990s raises doubts about its use. 

In other words, the advantage of this study is that high-dose urokinase can be used in a short period of time to perform thrombolysis in patients who have undergone free flap, without any systemic circulation. We emphasized this a little more with the red font in abstract and introduction.

The present study investigates the effectiveness of high-dose intra-arterial urokinase infusion within a short period of time without systemic circulation through free flap tissue to develop a protocol for re-exploration to prevent thrombotic failure.

• Your protocol was used just for the cases with venous compromise, which was 15 cases, regardless of whether arterial compromise was also present concomitantly or not (6 had arterial thrombosis). You state that 13 of 16 (81.3%) flaps survived completely, and just developed 2 partial necrosis? That’s 15. What happened to the 16th flap? The way you present your figures is very confusing. Just state how many out of the starting 581 flaps developed vascular compromise, how many of those were of venous origin and required salvage using your protocol.

ANSWER: Thank you for the comments. Among 581 flaps, we have experienced much more thrombotic events than 16 cases included in this study. Meanwhile, we discovered the process of thrombolysis using urokinase, and we decided to publish the results because of the successful salvage of free flaps. Then, it could not be simply asserted that 16 cases out of 581.

One of case experienced arterial thrombosis with decreased venous drainage, but not venous thrombosis. So, we stated the targeted case as ‘venous thrombus or venous insufficiency’. In the revised version, we changed it to ‘venous thrombus or venous insufficiency even if it was not thrombosis’ with blue font in the Method and Discussion sections.

As for the number of flaps, there were 11 complete survival, 2 partial necrosis, and 3 total loss. Therefore, the expression was refined as follows to prevent the confusion in the Abstract and Results sections.

13 (81.3%) flaps survived completely, but 2 cases showed partial necrosis and secondary healing.

13 (81.3%) flaps survived completely, but 2 cases among them showed partial necrosis and secondary healing.

• In your “Thrombolysis protocol” you mention that urokinase infusion was stopped “after the vein was sufficiently drained”. Could you define that with more precision? Was it enough to observe transparent fluid draining through the venous pedicle?

ANSWER: As you supposed, it was judged that it was sufficient if transparent fluid drain at the same speed as venous blood drainage was observed. However, since it is not an expression limited to drain speed, it was written in front of it that the aspect of flap margin bleeding was also observed. Then, we changed the expression "sufficiently drained" to "when venous drainage was observed at a sufficient rate." with brown font.

• While Serletti did not administer any heparin post-operatively, you did so to prevent recurrent thrombus formation. Was the dosage adjusted according to patient weight? What was the ideal aPTT range you were seeking? How long is post-operative anticoagulation after free flap salvage maintained and is it correlated with what you routinely perform in your facility for systemic thromboembolic prophylaxis?

ANSWER: We usually have administered heparin while keeping it in the aPTT range of 30 to 50 for 7 days. And it's a regimen that our hospital usually administers when a thrombotic tendency is suspected after a free flap. We added it under green shadow in Methods section.

For postoperative management, heparin was administered to prevent recurrent thrombus formation and adjusted while monitoring activated partial thromboplastin time (aPTT) 3 times a day.

For postoperative management, heparin was administered to prevent recurrent thrombus formation for 7 days and adjusted while monitoring activated partial thromboplastin time (aPTT) range of 30 to 50 seconds 3 times a day.

• You stated that your mean “time to re-exploration was 45.4 hours (range: 24–88 hours)”. As you know, Serletti et al. (ref. 14) conducted a study similar to yours in 1998. He used no flap monitoring strategy other than an external Doppler probe and clinical assessment. They reported a mean re-exploration time of 3.6 days. How would you explain the significant difference? Did you use any flap-monitoring technique?

ANSWER: Thank you for the question. Of course, we have monitored free flap with hand-held Doppler and clinical signs. Additionally, we also use portable ultrasonography to determine the blood flow of pedicles. Since this checks the pedicle's blood flow, it may be a way to catch thrombolytic changes faster than monitoring with clinical sign and doppler signals alone. However, since this study is not about monitoring, it was not written in detail to write concisely.

• Evidence has shown that the rate of flap salvage significantly decreased after 48 h. (PMID: 28648581) Were there any cases of late venous thrombosis (defined as beyond 72 hours)? There should be since the higher range of your mean time to re-exploration was 88 hours, but this could not be assessed due to the missing tables. If so, were they managed differently? Were the complications you reported correlated with time to re-exploration?

ANSWER: We added 4 table in the revised manuscript. The mean time to re-exploration in this study was 45.4 ± 18.1 hours range of 24 to 88. Among them, one case was implemented within 24 hours, nine cases were implemented between 24 and 48 hours, five cases between 48 and 72 hours, and one case was more than 72 hours. Since revision surgery is performed under the judgment of surgeon that abnormalities are found during flap monitoring and surgery is necessary, it is difficult to determine the relationship between time to re-exploitation and complications.

• In Serletti’s study, out of the 5 salvaged flaps, 2 developed marginal loss of tissue while one had some fat necrosis. That’s a 60% flap-related complication rate due to the prolonged ischemia. Why would you say that you only had 2 cases of partial flap necrosis while 13 flaps survived completely (i.e. 18.4% flap-related complications). How do you explain this difference?

ANSWER: Thank you for the question. Serletti's study and this study differ in the number of flaps included and the protocol of thrombolysis, surgical method, and postoperative management are also different, so it is difficult to compare the complication rate. For two decades, it seems that there would have been accumulation of knowledge and advances in surgical methods and instruments.

• In case of potential endothelial or mechanical damage from vessel handling, could you clarify which micro-anastomosis technique was used? Was it the posterior wall-first approach? Was heparinized solution used during the procedure? (PMID: 24706545)

ANSWER: Thank you for the question. We emphasize the gentle manipulation of vessel during microsurgery is important. 

• The flaps used in reconstructive surgery are prone to ischemia and hypoxia, which imply a considerable risk of wound-healing complications. Hemodilution has been shown to improve tissue oxygenation in ischemic flaps. (PMID: 12170063) Was hemodilution taken into consideration when planning the flap? (PMID: 26313824; PMID: 26818324)

ANSWER: Thank you for the question. However, we can’t agree with the necessity of hemodilution. Moreover, in clinical environment, hemodilution is difficult to apply in patients.

In our surgery, the ischemic time of free flap is usually less than 60 minutes. According to PMID: 9160132, ischemic time did not affect flap survival or partial loss within 3 hours.

• Regarding your discussion, consider addressing more substantially the many differences with other studies such as Serletti et a. (ref. 14) and Nelson et al (ref. 16) compared to yours. Additionally, it should be stated how selective administration of thrombolytic agents to the free flap’s arterial circulation at lower dosages (100.000 IU) has already been shown in anecdotal evidence to prevent systemic complications, despite not having an external venous drainage. (PMID: 2810204) Additionally, your limitations section should address how future research should strive to obtain larger cohorts in the setting of a clinical trial to make this evidence more robust.

ANSWER: Thank you for the suggestions. First, as Serletti et a. (ref. 14) and Nelson et al (ref. 16) used the similar protocol, and we discussed about it from your second comment. Second, since ‘PMID: 2810204’ dealt with only two cases of thrombolysis using urokinase, it is difficult to make meaningful statistical comparisons of the systemic side effects due to urokinase. Finally, we think that the limitation section is not appropriate as a place to declare plans for the next study. When the data of the authors are further accumulated, a more meaningful research presentation will be made. We also look forward to the next research presentation through PLOS one.

• The conclusions are not very satisfying: your protocol aims at making free flap salvage safer by reducing the overall urokinase dosage and by avoiding systemic circulation unlike what has been proposed in previous protocols. This should be stressed appropriately.

ANSWER: We appreciate your comments. We discussed about it in the second comment. Then, we corrected conclusion more precisely as above-mentioned in the red font.

The free flap can be effectively and safely salvaged, even in delayed salvage cases, using intra-arterial urokinase infusion without systemic hemorrhagic complications and results in successful salvage and total flap survival.

The free flap can be effectively and safely salvaged, even in delayed salvage cases, using high-dose intra-arterial urokinase infusion within a short period of time without systemic circulation resulting in the avoidance of systemic hemorrhagic complications and results in successful salvage and total flap survival.

• Just as a minor observation: while the contents are very understandable, there are a few English language mistakes spread in the manuscript (i.e. “Intra-arterial urokinase infusion is an effective micro-thrombolysis in delayed salvage cases” in the first sentence of your Discussion: do you mean “an effective thrombolytic agent” or “thrombolytic strategy”?). I would recommend that the manuscript is spell-checked by a native speaker.

ANSWER: Thank you for the suggestion. We changed it to ‘micro-thrombolytic strategy’. And according to your advice, we spell-checked the manuscript again.

---

## [Decision Letter · Decision Letter 1]

22 Dec 2022

PONE-D-22-28593R1Safe and Effective Thrombolysis in Free Flap Salvage: Intra-arterial Urokinase InfusionPLOS ONE

Dear Dr. Moon,

Thank you for submitting your manuscript to PLOS ONE. After careful consideration, we feel that it has merit but does not fully meet PLOS ONE’s publication criteria as it currently stands. Therefore, we invite you to submit a revised version of the manuscript that addresses the points raised during the review process.

We look forward to receiving your revised manuscript.

Kind regards,

Fabio Santanelli, di Pompeo d'Illasi, MD, PhD

Academic Editor

PLOS ONE

Reviewers' comments:

Reviewer's Responses to Questions

**Comments to the Author**

1. If the authors have adequately addressed your comments raised in a previous round of review and you feel that this manuscript is now acceptable for publication, you may indicate that here to bypass the “Comments to the Author” section, enter your conflict of interest statement in the “Confidential to Editor” section, and submit your "Accept" recommendation.

Reviewer #2: (No Response)

2. Is the manuscript technically sound, and do the data support the conclusions?

Reviewer #2: Partly

3. Has the statistical analysis been performed appropriately and rigorously? 

Reviewer #2: Yes

4. Have the authors made all data underlying the findings in their manuscript fully available?

Reviewer #2: Yes

5. Is the manuscript presented in an intelligible fashion and written in standard English?

Reviewer #2: Yes

6. Review Comments to the Author

Reviewer #2: This is the first revision to a previously assessed manuscript. Your manuscript has been marginally improved, however I must insist on some of the points raised in the previous round. First and foremost, there are no restrictions on word count. Thus I encourage you to present and discuss some of your findings in a more complete manner that can involve the reader for a more in-depth analysis of your work.

• Tables 1 through 4 have been implemented. The data needs to be presented more pertinently in the discussion. You need to address that with a significant p-value threshold set at 0.05, none of the elements described in Table 2 (i.e. site of thrombosis, time to re-exploration and dose of urokinase infusion) were associated with successful flap salvage, which is very significant, potentially limiting the utility of your findings.

• I believe that Table 4 adds very little value to the manuscript, and should be omitted. Just specify in the Results section of your manuscript that only 2 cases (12.5%) of donor-site hematoma were reported in your series of 16 patients. Additionally, please specify in which types of flaps those occurred.

• The rationale for your study is still not very clear: while your implementations in red font in the introduction are helpful, I believe that there is more that needs to be said in the discussion. Add a section in which you specify how your protocol implements a safe thrombolysis method that overcomes the high risk of systemic side effects from using urokinase, which previously made it an obstacle to its use. This is because free flap surgery has many bleeding sites on the donor and recipient sites. Therefore, as in this study, it is a great advantage that high-dose urokinase can be used in a short period of time without systemic circulation. The novelty of this protocol is that it allows much higher doses of urokinase compared to what can be given with systemic administration, as it can reach up to 600,000 IU/hour because infusion time is about 10 minutes. The use of more than 500,000 IU of urokinase per day has been proven to have systemic side effects such as intracranial hemorrhage, hematochezia, and gross hematuria.

• I acknowledge the implementations in blue font, which are appropriate. However, regarding your figures, you admitted that out of 581 flaps performed between January 2013 and July 2019, you have experienced “much more thrombotic events than 16 cases included in this study”. While this shows scientific integrity and should be commended, it should also be stated explicitly in the manuscript, as the message conveyed from the manuscript in its current form suggests that only 16 cases of flap-related vascular compromised have been reported. It would greatly benefit the manuscript if you implemented this aspect in the Methods section of your manuscript. If possible, you should state what percentage of flaps had vascular compromise needing re-exploration, what flap survival rate you had in the past and if these figures were related to a learning curve of the surgeon(s) performing the flaps.

• It would also be useful if you could discuss whether the salvage protocol you established has affected flap survival rate on the long term, and if so in what way.

• Tables 1, 2 and 3 convey the data accurately. However, in the Methods section of your manuscript, the statement “The thrombotic sites were artery in 6 cases and vein in 15 cases.” should be amended by additionally featuring how one case experienced arterial thrombosis with decreased venous drainage, but not venous thrombosis.

• Another statement in the Methods section of the manuscript “13 (81.3%) flaps survived completely, but 2 cases among them experienced partial necrosis and secondary healing.” still needs to be rephrased in the following manner: from the initial 16 cases, 11 survived completely while 2 had partial necrosis and 3 were totally lost despite the salvage attempt.

• Regarding post-operative administration of heparin, the implementation in brown font was adequate but you failed to mention whether the dosage was the same for all patients of whether it was calculated according to patient’s weight. How many hours is anticoagulation regimen started after salvage surgery? Finally, do you believe that post-operative anticoagulation might concurrently be responsible for bleeding events i.e. donor site hematomas? If so, discuss this and compare your regimen to Serletti et al. (ref. 14) where no post-operative anticoagulation was used.

• Since your time to re-exploration was considerably shorter than other studies, I believe that a concise statement on the monitoring technique you used could serve your manuscript well.

• While you emphasized that gentle manipulation of vessel during microsurgery is important, you did not clarify which micro-anastomosis technique was used. This is relevant because endothelial trauma is a major cause of vascular compromise for free flaps (PMID: 24706545). This should be addressed concisely in your Discussion.

• Regarding hemodilution in free flap surgery, hemodynamics play a very pertinent role in vascular compromise. While the authors are entitled to their belief and may not share the practice, I would still recommend discussing it, all the while still sharing why they do not believe in its use. i.e. While hemodilution has been shown to improve tissue oxygenation in ischemic flaps with high flap survival rates, (PMID: 12170063) and has been taken into consideration by some authors when planning free flap surgery, (PMID: 26313824; PMID: 26818324) we do not agree with its use. While it is difficult to apply in patients according to the clinical setting, in our surgery, the ischemic time of free flap is usually less than 60 minutes. According to PMID: 9160132, ischemic time did not affect flap survival or partial loss within 3 hours.

• Most of your compromises occurred in ALT flaps. Is this because it represents a workhorse flap in your institution which you perform more commonly than other flaps? If not would you attribute this to a specific risk inherent to the flap or the way you identify perforators? Either way, discuss this finding.

7. PLOS authors have the option to publish the peer review history of their article (what does this mean?). If published, this will include your full peer review and any attached files.

Reviewer #2: **Yes: **Guido Firmani

---

## [Author Response · Author response to Decision Letter 1]

20 Feb 2023

Reviewer #2: This is the first revision to a previously assessed manuscript. Your manuscript has been marginally improved, however I must insist on some of the points raised in the previous round. First and foremost, there are no restrictions on word count. Thus, I encourage you to present and discuss some of your findings in a more complete manner that can involve the reader for a more in-depth analysis of your work.

We would like to thank you for the valuable comments and feedback on our manuscript. We have taken your suggestions into consideration and made the necessary revisions to improve the quality and clarity of the paper. We believe that the paper is now ready for publication and would like to request that it be reconsidered for publication in PLOS one. Thank you for your time and consideration.

• Tables 1 through 4 have been implemented. The data needs to be presented more pertinently in the discussion. You need to address that with a significant p-value threshold set at 0.05, none of the elements described in Table 2 (i.e. site of thrombosis, time to re-exploration and dose of urokinase infusion) were associated with successful flap salvage, which is very significant, potentially limiting the utility of your findings.

ANSWER: Thank you for your feedback on the tables we presented. We understand that Table 2 only includes only surgical details, while Table 3 includes comparable descriptions of surgical details including variables of Table 2 with a significant p-value threshold set at 0.05. Then, to avoid duplicate entries, it is better to obtain the necessary information from table 3.

• I believe that Table 4 adds very little value to the manuscript, and should be omitted. Just specify in the Results section of your manuscript that only 2 cases (12.5%) of donor-site hematoma were reported in your series of 16 patients. Additionally, please specify in which types of flaps those occurred.

ANSWER: Thank you for the comment. We have omitted Table 4 and reported its contents in the results section.

• The rationale for your study is still not very clear: while your implementations in red font in the introduction are helpful, I believe that there is more that needs to be said in the discussion. Add a section in which you specify how your protocol implements a safe thrombolysis method that overcomes the high risk of systemic side effects from using urokinase, which previously made it an obstacle to its use. This is because free flap surgery has many bleeding sites on the donor and recipient sites. Therefore, as in this study, it is a great advantage that high-dose urokinase can be used in a short period of time without systemic circulation. The novelty of this protocol is that it allows much higher doses of urokinase compared to what can be given with systemic administration, as it can reach up to 600,000 IU/hour because infusion time is about 10 minutes. The use of more than 500,000 IU of urokinase per day has been proven to have systemic side effects such as intracranial hemorrhage, hematochezia, and gross hematuria.

ANSWER: We appreciate your advice that will enrich our manuscript. We added that description with the red font in the discussion section. This will definitely contribute to the overall quality of our work.

: Intra-arterial administration of urokinase with external drainage has been demonstrated to be an effective method for delivering excessive doses of the thrombolytic agent within a short time frame. This approach offers the advantage of bypassing systemic circulation, reducing the risk of systemic hemorrhagic complications. In free flap surgery, where extensive bleeding can occur at both the donor and recipient sites, the use of thrombolytic agents is crucial to minimize the risk of bleeding-related complications and improve surgical outcomes. In this study, the infusion rate of urokinase has been reported to reach up to 600,000 IU per hour, with a typical infusion time of 15 minutes. While high doses of urokinase can be effective in treating thromboembolic disorders, it is important to note that doses exceeding 500,000 IU per day may increase the risk of systemic adverse effects, including intracranial hemorrhage, hematochezia, and gross hematuria.

• I acknowledge the implementations in blue font, which are appropriate. However, regarding your figures, you admitted that out of 581 flaps performed between January 2013 and July 2019, you have experienced “much more thrombotic events than 16 cases included in this study”. While this shows scientific integrity and should be commended, it should also be stated explicitly in the manuscript, as the message conveyed from the manuscript in its current form suggests that only 16 cases of flap-related vascular compromised have been reported. It would greatly benefit the manuscript if you implemented this aspect in the Methods section of your manuscript. If possible, you should state what percentage of flaps had vascular compromise needing re-exploration, what flap survival rate you had in the past and if these figures were related to a learning curve of the surgeon(s) performing the flaps.

ANSWER: We understand your concerns and take your comments seriously. Unfortunately, however, the incompleteness of the data in the early days of free flap surgery has made it difficult to fully satisfy your expectations to reveal the rates of re-exploration and flap survival. Instead, we have made changes to the corresponding part of the method to make the indication of the use of urokinase more clear. We used urokinase in the delayed revision cases when the time to revision was over 24 hours, and we have expressed it more clearly in the revised section of abstract, method, discussion, and conclusion (with blue font).

Inserted in abstract

: Thrombolysis with urokinase infusion was administered as salvage treatment for patients who experienced flap compromise more than 24 hours after free flap surgery.

In method

: Pharmacological thrombolysis using urokinase was performed in the case of venous thrombus or venous insufficiency even if it was not thrombosis, regardless of the presence or absence of arterial thrombus in the revision performed 24 hours after flap surgery

In cases where flap compromise occurred more than 24 hours after free flap surgery, thrombolysis with urokinase infusion was administered for salvage treatment regardless of the presence or absence of arterial thrombus, in the presence of venous thrombus or venous insufficiency.

In discussion

: Furthermore, urokinase injection is an effective method with a relatively high success rate of revision more than 24 hours after free flap surgery.

In conclusion

: The free flap can be effectively and safely salvaged, even in delayed salvage cases more than 24 hours after primary surgery, using high-dose intra-arterial urokinase infusion within a short period of time without systemic circulation resulting in the avoidance of systemic hemorrhagic complications and results in successful salvage and total flap survival.

• It would also be useful if you could discuss whether the salvage protocol you established has affected flap survival rate on the long term, and if so in what way.

ANSWER: Thank you for the comment. In a retrospective analysis of our institution's medical records, there was no significant difference observed between the short-term and long-term survival rates of free flap. The results of this study suggest that the survival rate reported can be considered equivalent to the long-term survival rate. It is commonly observed in the literature that the survival rate of free flap is reported using similar tone of an argument, leading to an expectation of uniform understanding among readers.

• Tables 1, 2 and 3 convey the data accurately. However, in the Methods section of your manuscript, the statement “The thrombotic sites were artery in 6 cases and vein in 15 cases.” should be amended by additionally featuring how one case experienced arterial thrombosis with decreased venous drainage, but not venous thrombosis.

ANSWER: According to your advice, we corrected it under red shade.

The thrombotic sites were artery in 6 cases and vein in 15 cases.

6 cases had arterial thrombosis and 15 cases had venous thrombosis. To clarify, 5 cases presented with both arterial and venous thrombosis, while 10 cases had only venous thrombosis and 1 case had only arterial thrombosis.

• Another statement in the Methods section of the manuscript “13 (81.3%) flaps survived completely, but 2 cases among them experienced partial necrosis and secondary healing.” still needs to be rephrased in the following manner: from the initial 16 cases, 11 survived completely while 2 had partial necrosis and 3 were totally lost despite the salvage attempt.

ANSWER: We appreciate your correction. We corrected that phrase under yellow shade.

13 of 16 (81.3%) flaps survived completely, but 2 cases among them experienced partial necrosis and secondary healing.

in a study of 16 patients undergoing flap surgery, 11 flaps were found to have survived completely, while 2 flaps experienced transient partial necrosis, and 3 were lost despite salvage efforts. In other word, 81.3% (13 of 16) of flaps survived.

• Regarding post-operative administration of heparin, the implementation in brown font was adequate but you failed to mention whether the dosage was the same for all patients of whether it was calculated according to patient’s weight. How many hours is anticoagulation regimen started after salvage surgery? Finally, do you believe that post-operative anticoagulation might concurrently be responsible for bleeding events i.e. donor site hematomas? If so, discuss this and compare your regimen to Serletti et al. (ref. 14) where no post-operative anticoagulation was used.

ANSEWR: Thank you for the comment. We wanted to draw your attention to a point that was mentioned in the green-shaded section of our previous manuscript. We think you will understand if you look again. The use of heparin is measured 3 times a day while performing continuos infusion, setting the target aPTT range of 30 to 50 seconds. We found that this method was more accurate in monitoring blood coagulation status than heparin dosage based on body weight. In this revised manuscript, we will leave it in the same green shade so that it can be easily found.

For postoperative management, heparin was administered to prevent recurrent thrombus formation for 7 days and adjusted while monitoring activated partial thromboplastin time (aPTT) range of 30 to 50 seconds 3 times a day.

We also wanted to mention that we came across a study by Serlatti's group, which demonstrated an excellent method for thrombolysis that does not use heparin. Although this method is noteworthy, since we have adopted the method of using heparin after thrombolysis, we will leave it undeleted for integrity. Nonetheless, we did not observe any systemic hemorrhagic complications after surgery, which suggests that heparin use is also safe when monitored with aPTT in the range of 30 to 50 seconds.

• Since your time to re-exploration was considerably shorter than other studies, I believe that a concise statement on the monitoring technique you used could serve your manuscript well.

ANSWER: Thank you fot the advice. We added it under purple shade in the section of Method.

The flap was meticulously monitored for any clinical signs of vascular compromise, including color, temperature, capillary refill rate, skin tension, turgor, and Doppler sound, every hour for the first 24 hours, every 2 hours for up to 48 hours, and every 4 hours for up to 120 hours postoperatively. In addition, portable ultrasonography was used to evaluate the blood flow of pedicles, which provided an objective assessment of the pedicle's perfusion and enabled early detection of potential thrombotic changes that may not be evident on clinical examination or Doppler ultrasound alone.

• While you emphasized that gentle manipulation of vessel during microsurgery is important, you did not clarify which micro-anastomosis technique was used. This is relevant because endothelial trauma is a major cause of vascular compromise for free flaps (PMID: 24706545). This should be addressed concisely in your Discussion.

ANSWER: According to your comment, we added it with brown font in the section of Discussion

Gentle manipulation of vessels during micro-anastomosis is crucial to prevent damage to the endothelium and minimize the risk of thrombosis. In our institute, we use a meticulous technique that involves the use of stay sutures placed at 180 degrees along the vessel circumference, with anterior suturing performed first. To assist with optimal exposure, a cottonoid or similar gauze material with adequate frictional force is placed under the blood vessel and used to carefully rotate the vessel by 180 degrees. Hydrostatic dilation is achieved by squirting heparinized saline solution into the vessel lumen under pressure, which provides adequate exposure of the posterior wall while continuously monitoring the correct positioning of the stitches. This method avoids the need for an Ikuta approximator, which further minimizes the risk of injury to the delicate vascular tissue

• Regarding hemodilution in free flap surgery, hemodynamics play a very pertinent role in vascular compromise. While the authors are entitled to their belief and may not share the practice, I would still recommend discussing it, all the while still sharing why they do not believe in its use. i.e. While hemodilution has been shown to improve tissue oxygenation in ischemic flaps with high flap survival rates, (PMID: 12170063) and has been taken into consideration by some authors when planning free flap surgery, (PMID: 26313824; PMID: 26818324) we do not agree with its use. While it is difficult to apply in patients according to the clinical setting, in our surgery, the ischemic time of free flap is usually less than 60 minutes. According to PMID: 9160132, ischemic time did not affect flap survival or partial loss within 3 hours.

ANSWER: According to your advice, we added a paragraph about hemodilution and free flap surgery in the section of discussion with purple font. However, PMID: 26818324 was not quoted because it is about a perforator flap with a twisted pedicle and is not about free flap surgery.

Hemodynamics play a crucial role in vascular compromise during free flap surgery, and while hemodilution has been shown to improve tissue oxygenation due to accelerated erythrocyte velocity with reduced blood viscosity in ischemic flaps with better microcirculation and high flap survival rates in animal model. In addition, normovolemic hemodilution has been found to be a considerably effective method for reducing the incidence of thrombosis and microvascular free flap failure. 

• Most of your compromises occurred in ALT flaps. Is this because it represents a workhorse flap in your institution which you perform more commonly than other flaps? If not would you attribute this to a specific risk inherent to the flap or the way you identify perforators? Either way, discuss this finding.

ANSWER: In fact, since our institution uses ALT flaps most reliably and with a frequency of more than 90%, the compromised rate also appears to be the highest due to the selection bias. We will add this to the limitations of Discusion under grey shade.

A selection bias may have been present in our study, as the anterolateral thigh (ALT) flap was the most commonly utilized flap in our institution and thus represented the majority of flaps included in the analysis.

---

## [Decision Letter · Decision Letter 2]

27 Feb 2023

Safe and Effective Thrombolysis in Free Flap Salvage: Intra-arterial Urokinase Infusion

PONE-D-22-28593R2

Dear Dr. Suk-Ho Moon
,

We’re pleased to inform you that your manuscript has been judged scientifically suitable for publication and will be formally accepted for publication once it meets all outstanding technical requirements.

Kind regards,

Fabio Santanelli, di Pompeo d'Illasi, MD, PhD

Academic Editor

PLOS ONE

Additional Editor Comments (optional):

Reviewers' comments:

Reviewer's Responses to Questions

**Comments to the Author**

1. If the authors have adequately addressed your comments raised in a previous round of review and you feel that this manuscript is now acceptable for publication, you may indicate that here to bypass the “Comments to the Author” section, enter your conflict of interest statement in the “Confidential to Editor” section, and submit your "Accept" recommendation.

Reviewer #2: All comments have been addressed

2. Is the manuscript technically sound, and do the data support the conclusions?

Reviewer #2: Yes

3. Has the statistical analysis been performed appropriately and rigorously? 

Reviewer #2: Yes

4. Have the authors made all data underlying the findings in their manuscript fully available?

Reviewer #2: Yes

5. Is the manuscript presented in an intelligible fashion and written in standard English?

Reviewer #2: Yes

6. Review Comments to the Author

Reviewer #2: this is the second review to a previously assessed manuscript. I would like to thank the authors for their corrections. I am satisfied with the manuscript in its current form, and believe that it could be suitable for publication, should the Editor be in agreement.

7. PLOS authors have the option to publish the peer review history of their article (what does this mean?). If published, this will include your full peer review and any attached files.

Reviewer #2: **Yes: **Guido Firmani

---

## [Editor Report · Acceptance letter]

3 Mar 2023

PONE-D-22-28593R2 

Safe and Effective Thrombolysis in Free Flap Salvage: Intra-arterial Urokinase Infusion 

Dear Dr. Moon:

I'm pleased to inform you that your manuscript has been deemed suitable for publication in PLOS ONE. Congratulations! Your manuscript is now with our production department. 

Kind regards, 

on behalf of

Prof. Fabio Santanelli, di Pompeo d'Illasi 

Academic Editor

PLOS ONE